# A flexible framework for minimal biomarker signature discovery from clinical omics studies without library size normalisation

**Daniel Rawlinson[1,2], Chenxi Zhou [3,4], Myrsini Kaforou[5], Kim-Anh Lê Cao[2], Lachlan J. M. Coin [1,6]\*, RAPIDS Study Group[¶]**

1 Department of Microbiology and Immunology, University of Melbourne at The Peter Doherty Institute for Infection and Immunity, Melbourne, Australia, 2 Melbourne Integrative Genomics, School of Mathematics and Statistics, University of Melbourne, Melbourne, Australia, 3 Department of Genetics, University of Cambridge, Cambridge, United Kingdom, 4 Wellcome Sanger Institute, Wellcome Genome Campus, Cambridge, United Kingdom, 5 Department of Infectious Disease, Faculty of Medicine, Imperial College London, London, United Kingdom, 6 Department of Clinical Pathology, University of Melbourne, Melbourne, Australia

¶ Other members of Rapid Paediatric Infection Diagnosis in Sepsis (RAPIDS) study group are listed in the Acknowledgments section.
* lachlan.coin@unimelb.edu.au

## Abstract

Application of transcriptomics, proteomics and metabolomics technologies to clinical cohorts has uncovered a variety of signatures for predicting disease. Many of these signatures require the full 'omics data for evaluation on unseen samples, either explicitly or implicitly through library size normalisation. Translation to low-cost point-of-care tests requires development of signatures which measure as few analytes as possible without relying on direct measurement of library size. To achieve this, we have developed a feature selection method (Forward Selection-Partial Least Squares) which generates minimal disease signatures from high-dimensional omics datasets with applicability to continuous, binary or multi-class outcomes. Through extensive benchmarking, we show that FS-PLS has comparable performance to commonly used signature discovery methods while delivering signatures which are an order of magnitude smaller. We show that FS-PLS can be used to select features predictive of library size, and that these features can be used to normalize unseen samples, meaning that the features in the complete model can be measured in isolation for making new predictions. By enabling discovery of small, high-performance signatures, FS-PLS addresses an important impediment for the further development of precision medical care.

## Author summary

High-throughput sequencing technologies are widely used in clinical studies to measure expression levels of many thousands of different types of molecules in order to develop improved models for predicting disease state and progression. However, low-cost diagnostic assays can only measure a handful of molecules. We have developed a framework,

**Data availability statement:** This manuscript uses publicly available data. The accessions are provided in the manuscript. Additionally, pre-pared data tables for the recreation of results are deposited at our repository: https://github.com/dn-ra/FSPLS-publication-repo.

**Funding:** This work was supported by the National Health and Medical Research Council (GNT1195743 to LC) and the Medical Research Future Fund (GHFM76734 to LC). The funders had no role in study design, data collection and analysis, decision to publish, or preparation of the manuscript.

**Competing interests:** The authors have declared that no competing interests exist.

called FS-PLS, for identifying minimal sets of biomarkers for predicting disease state. Here we show that the minimal gene signatures retain almost all of the accuracy of larger models. Additionally, translation of models developed from high-throughput datasets typically require correction for the total number of molecules sequenced, referred to as library size. We show that FS-PLS can also be used to obtain reliable predictions of library size from measurement of only a few molecules.

## Introduction

The goal of providing accurate disease diagnosis from high-throughput 'omics assays is being actively pursued by biologists, computational scientists, and diagnostics companies. The omics assays in question - transcriptomics or proteomics for instance - are rich in information and typically capture many thousands of features to describe the biological state of the patient or donor. Indeed, the number of features measured usually exceeds the number of samples collected in a study by several fold, which presents difficulties for optimal fitting of coefficients to the features included in the model [1].

A machine learning (ML) approach to diagnostics development seeks to take the high-dimensional data generated with an omics assay and generate a mapping from the input feature space to an output probability or prediction of disease status. As computing power has accelerated and complex ML models have become more accessible, much of the application of ML to biology has embraced the full feature set to train models and provide new predictions e.g., [2,3]. However, feature selection, or the refinement of the feature set to a smaller number of the most relevant molecules, is still of utility in the clinical domain. Firstly, feature selection offers interpretability as to what aspects of biology are important for the predicted diagnosis [4]. Secondly, it addresses the problems associated with generating a model using an over-whelming number of features against comparatively few samples [5]. And thirdly, a minimal feature set adds the possibility for conversion of the 'omics test into a readily available manu-factured test that is cheaper and faster than a complete 'omics assay [6]. The development of simplified testing procedures is of particular benefit for resource-constrained settings where complex laboratory procedures are impracticably onerous [7].

Existing methods for feature selection fall into three broad categories of wrapper, filter, or embedded approaches [8] Wrapper-based methods select individual features sequentially and fit a model after each iteration, thus allowing the algorithm to decide when a sufficient number of features have been included. Forward selection, backward selection, and recursive feature elimination all fit into this category. Filter-based methods determine a univariate measure of relevance for each candidate feature (e.g., $R^2$, Spearman's $\rho$), and apply a threshold cut-off to include just the most relevant features into a model. Embedded approaches unite the feature selection and model building steps such that an objective function can be evalu-ated, and an optimal model can be analytically derived [8]. Additionally, projection-based methods have seen extensive use in the biological domain due to the very high dimensional structure of 'omics datasets. While projection-based methods are helpful for managing high-dimensionality, these methods collapse the dimensionality down into orthogonal, latent components rather than selecting individual features for modelling, so further steps need to be taken to restrict the actual number of features [e.g., 9].

Embedded feature selection methods using regularisation are used extensively in research on signature generation [e.g., 10]. Occupying this category are LASSO [11] and Elastic-Net [12], both of which solve linear models of X regressed on y with a penalty imposed on the L1 norm of the maximum likelihood coefficients. The two differ in that LASSO imposes a penalty

strictly on the L1 norm, whereas Elastic-Net uses an additional mixing parameter *alpha* to modulate the proportion of the penalty meted on the L1 and L2 norms. Consequently, Elastic-Net generally produces models with more non-zero feature coefficients, but these have lower magnitude owing to the influence of the L2 portion of the penalty in an effort to limit overfitting. Many modified regularisation methods have been employed in signature discovery, such as Adaptive LASSO [13], Relaxed LASSO [14], and Bayesian formulations [15]. However, the original LASSO and Elastic-Net designs remain widely used.

Minimum Redundancy Maximum Relevance (mRMR) is a filter-based method that is popular for biological signature generation because it reduces collinearity in the final feature set. mRMR ranks the candidate features by balancing two criteria: rewarding mutual information shared between the feature and the outcome variable, and penalising mutual information shared between each feature and all features preceding it [16]. The modeller can then select the number of minimally redundant features that is desired for modelling with some other function [17].

Whatever method is used for deriving a small biological signature from compositional (rather than absolute) count data, application of the signature to unseen data still requires a full 'omics dataset for normalisation. Signatures requiring normalisation are not truly small as all features must be measured to resolve the compositional constraint of the key features in the signature. This has typically been addressed by the somewhat ad-hoc use of house-keeping genes for count normalisation, yet the optimal strategy for normalisation remains a significant impediment to the translation of biological signatures into minimal diagnostic tools.

In this work we describe Forward Selection – Partial Least Squares (FS-PLS), a novel and flexible method for feature selection from omics data that delivers small signatures for prediction of disease state in binary and multinomial classification problems. FS-PLS combines the dimensional reduction and component orthogonality of projection-based modelling methods (PLS) with the sparsity and clinical interpretability of a wrapper-based method (FS).

We demonstrate that FS-PLS generates sparse signatures with little deterioration of performance when compared with more dense models, and with improved performance in comparison with other sparse feature selection methods. Additionally, by applying FS-PLS to select features that predict library size, we show that classification models can incorporate normalisation features that do away with the need for a true library size and require only the minimal feature set for prediction of new samples. Further, the derived signatures suffer no significant loss in accuracy compared with conventionally normalised samples. In this way FS-PLS provides a unified solution to identification of complete signatures of disease, which is applicable to any compositionally-constituted 'omics biomarker discovery problem.

FS-PLS has already been used to discover biomarkers from gene expression data [18–20]. Here our aim is to evaluate its effectiveness against the backdrop of widely used signature generation methods. We assess FS-PLS's performance against the embedded methods LASSO and Elastic-Net, and the filter method mRMR (minimum Redundancy-Maximum Relevance), which ranks features according to the amount of shared information with the outcome variable while penalising their shared information with other variables. We apply the methods to five distinct 'omics datasets: two microarray, two RNA-Seq, and one proteome (see *Methods*), which are comprised of three binary and two multi-class outcome types. Additionally, we demonstrate FS-PLS's advantages over standard forward selection approaches of Stagewise FS for binary and StepAIC for multinomial. Our comparison is focused on linear methods as these explicitly select or rank features for inclusion rather than inferring feature importance once a classifier has been made as with tree-based or neural network modelling. We follow our comparison of methods with an assessment of FS-PLS's performance with the incorporation of normalisation features on the two RNA-Seq datasets and with a separate validation dataset for one of these.

Ultimately, our work shows that FS-PLS is a robust method for feature selection from omics data that should be considered when researchers are seeking minimal signatures of disease state. FS-PLS is available in open access at https://github.com/lachlancoin/fspls.

## Methods

### Description of FS-PLS

FS-PLS is designed to ensure that each successive feature explains a portion of the outcome that is not already explained by the features that have been selected before it. It does this by identifying the direction in the data that is accounted for by a chosen feature, removing that direction from the data matrix, then making the remaining data available for consideration of the next most useful feature.

Denote $X$ a matrix of observations (m) with variables (p) and an outcome variable $y$, which can be binary, multinomial, or continuous. The matrix $X$ is centred so that the mean of each variable is 0, and the means of the variables are recorded as $\mu_1, \ldots, \mu_p$. The pseudocode below describes the main steps of the algorithm:

```
Input: Dataset matrix, Outcomes vector
1. Fit a univariate linear model for each column of dataset
   regressed on outcomes
2. Select feature with maximum likelihood
3. Fit coefficient with L2 shrinkage
4. If p-value of model with chi-squared test < threshold:
   a. Add values of selected variable to matrix Xk and coefficient
      to betas list
   b. Project X onto Xk, subtract projection from X to derive new
      orthogonal dataset matrix Rk
   c. Return to step one replacing original dataset matrix with Rk
5. Return variables and coefficients
```

Pseudocode of FS-PLS algorithm. See supplementary material for a more detailed version of pseudocode (S1 Info).

The FS-PLS algorithm proceeds in $k$ iterations, where at each iteration a new variable is selected and fitted until some stopping criteria is satisfied (described below). At the $k_{th}$ iteration we take $X^k$ to be the set of $k$ variables already selected from the data: $X^k = [x_{i_1}, \ldots, x_{i_k}]$, which at initialisation is a matrix of 0 columns. The Singular Value Decomposition (SVD) is applied to $X^k$ to obtain an orthonormal basis $U$ on the $k$-dimensional subspace of $X$, which is spanned by the $k$ selected variables of $X^k$.

The full-dimensional $X$ matrix is then projected onto $X_k$ with $P^k = UU^TX$ to determine the portion of each element of $X$ that is captured by the $k$-dimensional subspace. The original data $X$ is subsequently deflated with $P^k$ to obtain $R^k = X - P^k$, a new $mxp$ matrix whose columns are all orthogonal to the previously selected variables $X^k$. We note that while the columns of the previously selected variables are still present in $P^k$, they have zero variance.

From here, FS-PLS fits a univariate linear model for each $j$ of $R^k$ and chooses the one with the maximum log likelihood for addition into the feature set. In so doing, a new variable $j$ is selected that exhibits the greatest correlation with $y$ that is orthogonal to previous variables. For multinomial outcomes, the univariate modelling is achieved using the *nnet* package, which approximates a multinomial regression with a single hidden layer neural network [21].

Whereas standard forward selection might subtract the current estimate $\tilde{y}$ from $y$ to orthogonalize the remaining variation, FS-PLS instead borrows from PLS by deflating X. This is done so that at each successive iteration, the previously selected variables are completely

removed from the space of $X$ and have no chance of re-entering the model at a later stage. Moreover, removing the component of X that covaries with $X^k$ means that variables that correlate with $X^k$ are also precluded from entering the model in place of a previously selected $k$ at a later iteration.

Once a new variable is selected, FS-PLS re-fits its coefficient with L2 shrinkage applied as in ridge regression and the variable is added to $X^k$ for the next iteration of the algorithm. A chi-squared test is performed to determine the significance of the log-likelihood of the new variable against the null model (i.e., just the $y$ vector and the offset). FS-PLS stops adding variables once the p-value exceeds a threshold, or when a pre-defined maximum number of variables has been selected.

For prediction of new data, FS-PLS maintains the coefficients for each variable in its deflated space and applies the saved orthogonalization transformations at each variable. Model estimates are thus fixed once estimated at each variable selection step, which ensures that only the portion of each variable that is uncorrelated with previous variables is incorporated into the model.

## Datasets

**Microarray datasets.** The microarray dataset from Golub *et al.* [22] is a collection of bone marrow and peripheral blood samples from 72 patients with a diagnosis of either Acute Myeloid Leukemia (AML) or Acute Lymphoblastic Leukemia (ALL). RNA was hybridised to Affymetrix microarrays containing 6817 human gene probes to generate the data. The dataset has served as a classic benchmarking dataset in many published computational methods [23–25] and is available in processed and normalised form from https://www.kaggle.com/datasets/crawford/gene-expression.

Kaforou *et al.* [26] collected blood samples of 584 adults in Malawi or South Africa with active Tuberculosis infection, latent Tuberculosis infection, or other disease. Transcriptional microarray profiles were generated using HumanHT-12 v.4 expression Beadarrays (Illumina). For FS-PLS benchmarking, we retained only the active TB (n = 195) or latent TB samples (n = 167) and used these for binary classification. Data is available for download from NCBI's Gene Expression Omnibus under accession number GSE3725.

**RNA-seq datasets.** The RNA-seq data from Ng *et al.* [27] consists of 286 nasopharyngeal and 53 whole-blood samples from patients with COVID-19 and controls collected at University of California San Francisco. Sequencing of RNA samples was performed using the Illumina NovaSeq6000. For benchmarking, we used only nasopharyngeal samples and used the signature generation methods to classify COVID-19 (n = 138) from other acute viral respiratory illness (n = 120). RNA count tables were downloaded from the Gene Expression Omnibus under accession GSE163151.

The RAPIDS dataset was generated through the work of the Rapid Acute Paediatric Diagnosis of Infection in Suspected Sepsis study for the discovery of transcriptomic signatures for the diagnosis of bacterial vs viral sepsis in children [28]. Patients with suspected sepsis were recruited from Emergency Departments and Paediatric Intensive Care Units at four hospitals around Queensland, Australia. Whole blood samples were used for the study and split into a discovery cohort (n = 595) and a validation cohort (n = 316). RNA sequencing was performed on Illumina NovaSeq. Clinical diagnoses were confirmed and assigned to one of 6 classes: Definite Bacterial, Definite Viral, Probable Bacterial, Probable Viral, Non-infectious, or Unknown. For the benchmarking performed in this paper, Unknown cases were excluded and probable viral or bacterial cases were united with the definite diagnoses to construct a 3-class classification problem of Non-infectious vs Viral infection vs Bacterial infection.

Evaluation of models from RAPIDS data was performed using the validation cohort of the study.

**Proteomic dataset.** Álvez *et al.* [29] produced a dataset of 1477 samples and 1463 protein features using the OLink Explore PEA technology. The samples originated from patients with 12 different cancer types and healthy controls, but for the FS-PLS benchmarking we selected the samples of the four blood cancer types (AML, CLL, DLBCL, Myeloma) and merged all other cancer samples into a control class (CTRL) to construct a 5-class classification task for assessment of the feature selection methods.

## Design of benchmarking experiment

The experiment is conducted as illustrated in Fig 1. Samples from each dataset are divided into 5 folds to allow cross-validation of model performance. Training samples are normalised independently from the test set and are used to select discriminative features for class prediction. For feature selection we compare FS-PLS with LASSO, Elastic Net, and MRMR as commonly used signature generation methods, and with ordinary forward selection methods (stagewise forward selection for binary datasets, stepAIC for multi-class datasets) to demonstrate FS-PLS's improvement over its most related methods (see *Feature selection methods and settings*). The test set is then used to evaluate class performance. All methods use the same fold structure in the data to preserve comparability in results.

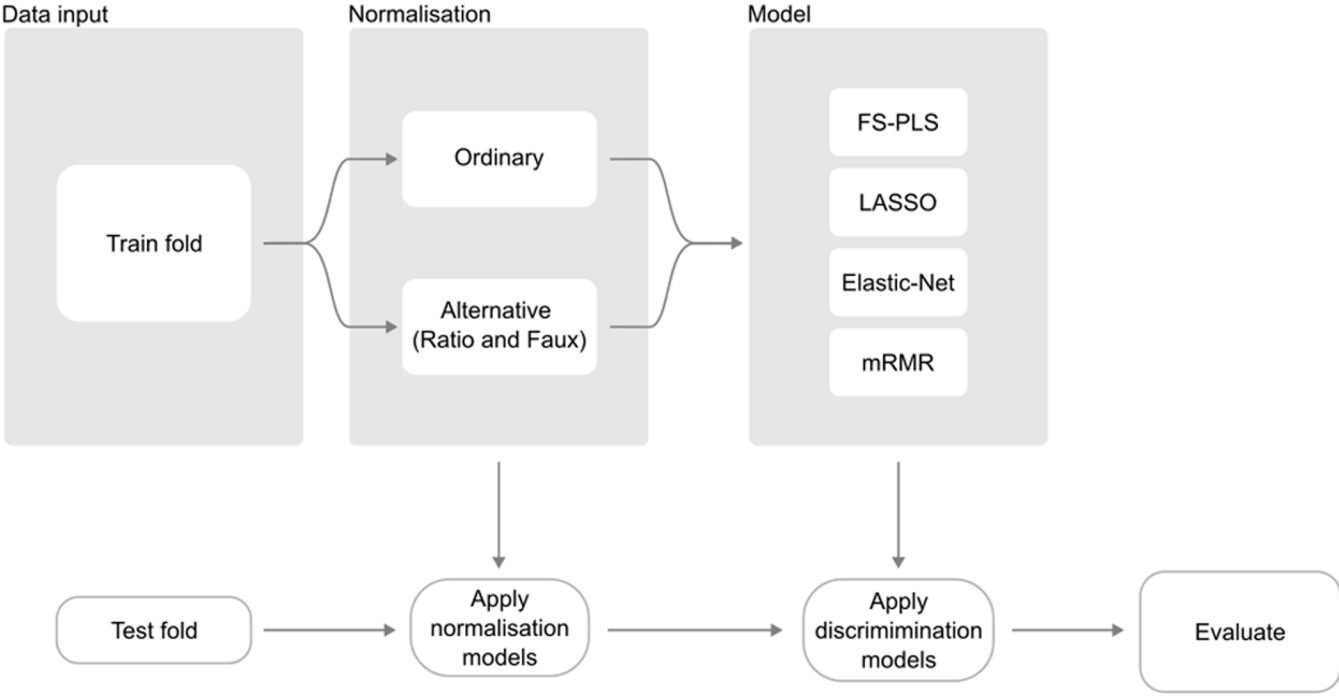

**Fig 1. Schematic of the experimental setup for benchmarking of FS-PLS against LASSO, Elastic-Net, and mRMR.** All datasets were normalised (ordinary normalisation) or, in the case of RNA-Seq data, modelled to reproduce log library size of samples with the fewest possible features and then normalised with the modelled solution (alternative normalisations). Following normalisation, we ran the feature discovery modelling to select features and learn a function to predict the outcome variable in question. Finally, models were evaluated by applying the various normalisations to test folds and generating new predictions using models trained on the respective normalisations. While Forward Stagewise and stepAIC methods were also tested in our benchmarking, there were not included in out alternative normalisation approach so are excluded from the schematic shown here.

Alongside the feature selection task, the training step for RNA-Seq data includes a *normalisation* feature discovery task where the log-library size of each sample is calculated and used as the continuous response variable for training of a sparse FS-PLS model to predict library size.

The computed normalisation model is fed into the discrimination feature discovery, where it contributes to two alternative normalisation solutions: "faux" normalisation based on predicted library size output by the normalisation model, and "ratio" normalisation where the original set of features is recast into a new feature space consisting of the log ratios of each feature and each normalising feature selected by the normalisation model.

The discrimination feature discovery task thus not only compares feature selection methods across folds but also compares prediction accuracy from three possible normalisation approaches: full library size normalisation, faux normalisation, and ratio normalisation. Discrimination models are trained on the differently normalised data, yielding 12 total models to be evaluated.

For evaluation of performance, the held-out test partition is once again normalised with the three different normalisation solutions: full library size normalisation, faux normalisation by imputing a library size using the pre-trained model, and ratio normalisation using the earlier selected features as denominators in the feature log ratios. Trained discrimination models are then applied to generate new predictions and are assessed against true classes according to the appropriate metric (see *Methods*). Crucially, the fold structure used for normalisation and discrimination training is the same, so that new predictions are made on samples that have not been seen by either model in the training stage.

## Data treatment

For each of the 5 datasets in the benchmarking, samples were split into 5 folds for cross validation of model accuracy, thus yielding 5 individual models per method per dataset with 4 folds used for training and one fold held out for testing. To shorten computation time, pre-filtering of features was applied by selecting the top 10000 most variable features (Kaforou, Ng, RAPIDS) prior to splitting data into folds. For microarray data, where auto-scaling of features was used (mean = 0, variance = 1), scaling was performed separately within training and test partitions to prevent data leakage. RNA-Seq data was log transformed but left unscaled to ensure new predictions could be generated on individual samples. Missing values in the Álvez dataset were imputed within each fold using the *knnImpute* method from the *preProcess* function in the *caret* package.

The samples within each fold were identical for each of the methods tested to preserve comparability of results across methods. Furthermore, samples in the training fold for each model were kept consistent across feature selection for classification and for normalisation tasks, so that final classification predictions were determined on samples that had not been seen by either the classification or the normalisation model.

To construct sparse models for imputing library size, true library sizes of the test samples were determined by summing feature counts per sample before any filtering. The vectors of library sizes were log-transformed and used as a gaussian outcome variable for which the feature selection and model generation methods were modelled using gene counts $X$.

When incorporating feature selection for library size normalisation, it is important to ensure that the selected features are stably expressed in the discovery and the evaluation datasets. Accordingly, we added another filtering layer on features input into library size modelling so that only features in common across the discovery and test datasets' 10000 most expressed features (.05 trimmed mean) were included. Gene counts were log-transformed but otherwise un-normalised and un-scaled before training. For the Ng data, the same 10000 most expressed used were used as candidates for normalisation and discrimination features. For the

RAPIDS data, discrimination features were further selected from genes in common across the 10000 most variable features in the discovery and validation sets.

## Feature selection methods and settings

We used the R packages *glmnet* [30] and *mRMRe* [31] for implementations of the alternative feature selection approaches. For LASSO and Elastic-Net (*glmnet)* an additional inner structure of 5 folds was used in the training data to select the strength of regularization (lambda). The implementation of MRMR (*mRMRe)* used in this study has no stopping criteria or model-fitting procedure and simply orders input features according to the MRMR loss function. Accordingly, we instructed MRMR to select the top *n* features equal to the number of features chosen by FS-PLS for that fold. We then estimated the coefficients of MRMR's chosen features with ridge regression in *glmnet* and 5-fold cross-validation to select the degree of shrinkage.

The standard forward selection (FS) procedure holds some substantial similarities with FS-PLS. Stagewise FS proceeds with a series of iterations, at each point selecting a new feature to include in the model and adding new weights in addition to adjusting the weights of the features already selected [32]. There are two primary differences between stagewise FS and FS-PLS. First, weights for selected features in FS-PLS are learned just once and then fixed in the deflated space with the aim of avoiding overfitting. Secondly, at each stage of the FS algorithm the current estimate is subtracted from the true class value to produce a residual outcome variable for selecting of the next feature, whereas FS-PLS employs the projection method of PLS to extract information already explained from the X matrix.

Given the similarities of the methods, it is prudent to evaluate FS-PLS in direct comparison with ordinary stagewise FS. We performed a head-to-head comparison of the two forward selection methods using the same 5-fold structure in each method for all datasets. For binary datasets, we used the *lars* package to generate stagewise FS models. However, stagewise FS is not applicable to multinomial data, as the current estimate at each stage cannot be subtracted from the outcome variable. To test an analogous method to stagewise FS in multinomial data, we used the *multinom* function from the *nnet* package in conjuction with *StepAIC* from the *MASS* package [21] to build a sparse multinomial model to a pre-selected number of features.

Loss functions were selected for each type of response variable that was to be predicted: Area Under the Curve (AUC) for binary outcomes, multinomial deviance for multi-class outcomes, and Mean-Squared Error (MSE) for the continuous library sizes in the normalisation modelling. The same metrics were used in final performance evaluation, except for multi-class models whose performance is presented in multiclass AUC, and sensitivity and specificity per class.

For each method in multi-class and continuous prediction tasks, we extracted two sets of coefficients: one which produced the minimum training error ('min'), and another which gave the maximum regularization strength that still produced training error within 1 standard error of the minimum ('1se'). In *glmnet*, this is performed automatically with the *cv.glmnet* function, but in FS-PLS we achieved the equivalent by directly selecting the number of features that gave the min and 1se training errors. We found that the set of features corresponding to the minimum training error generally performed best, so all reported classification results are from 'min' models apart from the normalisation feature selection models for which the '1se' models are used.

## Analysis environment and code availability

Analysis was performed in R version 4.2.1. Visualisation of results was achieved using *ggplot2* version 3.4.4, with assistance from *ROCR* [33] version 1.0.11 and *pROC* [34] version 1.18.4 for

deriving metrics, and *caret* [35] version 6.0.94 for partition splitting and missing value imputation in proteomic data.

FS-PLS is available for the community to use at https://github.com/lachlancoin/fspls. Scripts for creation and evaluation of results discussed in this paper are deposited at https://github.com/dn-ra/FSPLS-publication-repo.

## Results

### FS-PLS generates much smaller signatures compared with regularisation methods

We first assess FS-PLS in the setting in which the library size (or equivalent) information is available, for example when the test set is measured on the same platform as the training dataset.

We observed that FS-PLS generally achieved similar performance to LASSO and Elastic Net but with much fewer features selected for inclusion in the model (Fig 2). In two binary datasets, there was non-significant loss in performance from LASSO to FS-PLS according to mean difference in AUC (Kaforou: -0.02, Ng: -0.02) and a many-fold reduction in mean feature numbers (Ng: 4.6 (FS-PLS), 24.8 (LASSO); Kaforou: 5.6 (FS-PLS), 23.2 LASSO), S1 and S2 Data).

The Golub dataset diverged from the above results by demonstrating a sharp drop in FS-PLS performance in contrast with the other methods (Fig 2A). The results seen here are best explained by the small number of samples available in the dataset and will likely be recoverable by training on a larger dataset. We demonstrate this below in our comparison of FS-PLS with ordinary forward selection methods.

Moreover, as the sample size of the datasets increased, the reduction of feature numbers became more pronounced. Application of regularisation-based methods to multi-class problems led to several hundred features selected. In contrast, FS-PLS created a much smaller signature (mean # of features: 10 (RAPIDS dataset), 7.6 (Álvez dataset)) at a cost to multiclass AUC of 5-6 percentage points (Fig 2 and S3–S6 Data).

### FS-PLS outperforms MRMR for the equivalent number of selected features

Since MRMR was set to select the same number of features as FS-PLS, this represents a biased advantage for MRMR as it is borrowing the appropriate number of features learned by FS-PLS. Despite this, MRMR and FS-PLS selected different features to include in the model's set and delivered varying performance (Fig 2). Aside from the exception of the Golub dataset, as discussed above, FS-PLS outperforms MRMR binary classification models when trained and tested on common folds (+0.01, +0.02 mean AUC for Kaforou and Ng, respectively).

In multi-class datasets, FS-PLS outperforms again but is also more sensitive to underrepresented classes than MRMR (Fig 3). When weighting samples by their inverse class proportions for the RAPIDS dataset, performance of FS-PLS on all 3 classes actually deteriorates compared with no sample weights at all. Conversely, MRMR requires sample weights for 2 out of 3 classes to achieve best performance and is outperformed by FS-PLS in the aggregate.

Training FS-PLS without sample weights also performed poorly for the Álvez data. The control class is so dominant in number that the inclusion of any more than one variable leads to a deterioration of the training loss and the number of features selected never exceeds one. To bypass the model training difficulty, we trained an FS-PLS model for each fold through to a maximum number of 10 features regardless of training performance. As seen with the RAPIDS dataset, the unweighted FS-PLS Álvez models demonstrate superior or equal test AUC results for every class compared with the weighted MRMR models (Fig 3). The benefit

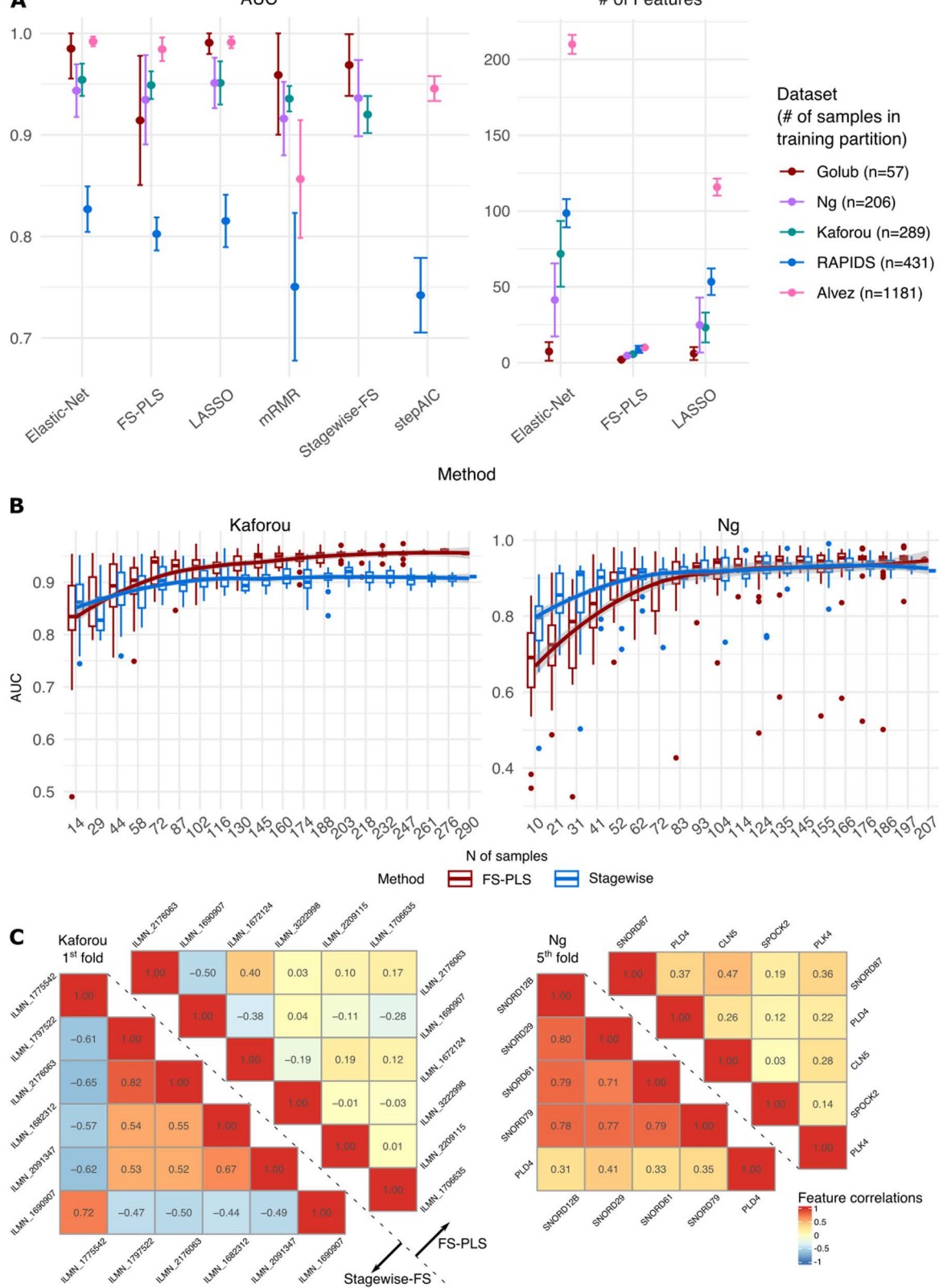

**Fig 2. Performance of examined methods on each dataset. A.** Test AUCs (left) and numbers of features selected (right) for all datasets and methods included in the study. For most datasets, FS-PLS performs similarly to LASSO and Elastic-Net with comparatively much fewer features. Mean and 95% confidence interval across each of 5-folds are shown. For multiclass datasets (RAPIDS,

Álvez), AUC value for each k-fold is derived from the average AUC across classes. Display of feature numbers selected excludes mRMR, Stagewise-FS, and stepAIC as these methods were instructed to select the same number of features as chosen by FS-PLS. Only one of Stagewise-FS or stepAIC are applied to each dataset, as they are the most appropriate forward selection strategies for binary or multinomial problems respectively. **B.** Performance of FS-PLS against ordinary Stagewise forward selection on Kaforou (left) and Ng (right) data at varying sample sizes to explain poor FS-PLS results in Golub dataset. Each box represents 20 repeated samplings of the data at the given sample size. Stagewise FS initially displays greater accuracy than FS-PLS at low sample sizes but is surpassed by it as more samples are included in the training set. **C.** Correlations of features selected by Stagewise FS (lower triangle) and FS-PLS (upper triangle) on Kaforou's 1st fold (left) and Ng's last fold (right) data. Diagonal dashed line is placed to emphasise the separation of the two methods. Features are presented in the order of selection from top-to-bottom and from left-to-right. FS-PLS consistently selects feature sets that have lower correlation than Stagewise FS. Plots from all folds are available in S4 and S5 Figs.

of sample weighting for mRMR models is unclear in the AUC results when compared with unweighted models. However, the meagre sensitivity of the LYMPH class (0.0) without weights is justification of adopting the weighted model as superior. Indeed, all alternative models suffer from meagre sensitivity for one class in both multinomial datasets when sample weights are unused.

Given FS-PLS's sensitivity to low proportion classes in the absence of weighting, it is unclear whether a more optimal strategy than the one used here may further improve performance. Nevertheless, the results of feature selection on multi-class datasets reported throughout this benchmarking take the unweighted FS-PLS models as compared with weighted models from the alternative methods.

In addition to comparing methods using AUC, sensitivity, and specificity, we implemented a Wasserstein metric to quantify the separation in cumulative predicted probabilities for each binary comparison in the multinomial classification problem (see *Methods)*. Visualising the divergence of probabilities provides an assessment of prediction performance that accounts for the clearance each method gives to candidate classes that is not visible in AUC calculations.

The difference in case vs control cumulative probability curves for FS-PLS is much higher than for MRMR, indicating a relatively greater degree of confidence in the probabilities being assigned (Fig 3). Indeed, the unweighted FS-PLS model on the RAPIDS data exhibits a greater separation in probabilities for each class than for all other weighted methods (S1 Fig). Altogether, these results indicate that FS-PLS is a method that delivers high-performance discrimination models with relatively few features chosen.

## FS-PLS performance relative to forward selection on binary datasets depends on sample size

As discussed previously, FS-PLS exhibits poor performance on the relatively simple Golub binary dataset, and so stagewise FS has superior accuracy on test partitions by an average of +0.055 AUC (Fig 2). In other binary datasets, however, FS-PLS either matches FS's performance (mean AUC change Ng +0.00) or surpasses it (Kaforou +0.03) using the same number of features (S7 Data).

We hypothesised that the poor performance of FS-PLS models from the Golub data are a consequence of the small number of samples used in training these (n = 57 per model). We trialled the theory by performing a sampling experiment on the Kaforou microarray and Ng RNA-Seq datasets. The data were split into 80%-20% train-test partitions. Random sampling of the observations in the train partition was performed at increments of 5% between 5% and 100% of the total partition size. Sampling was repeated 20 times within each sample size. FS and FS-PLS models were trained on the sampled datasets, ultimately yielding 400 models (20

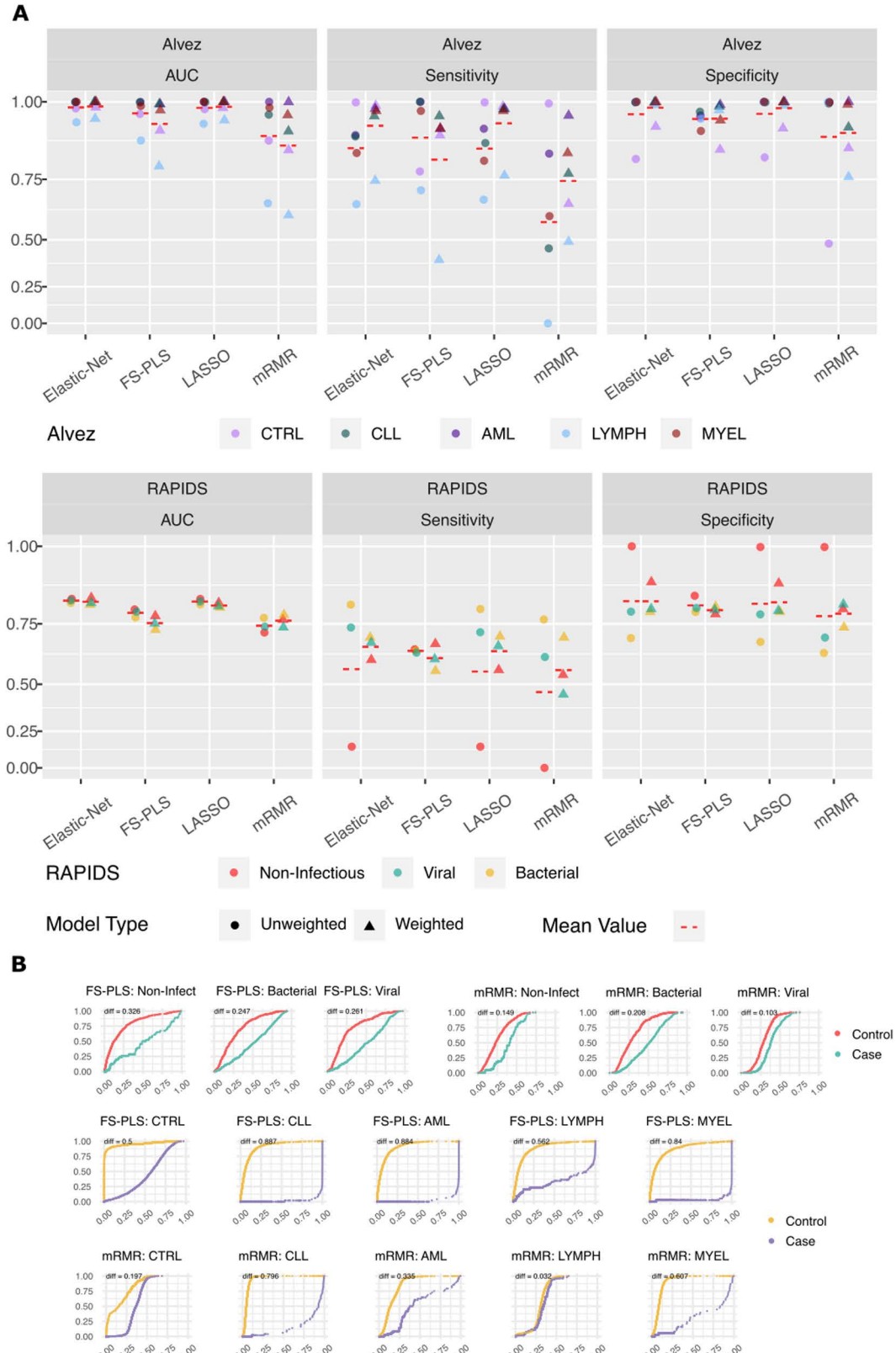

**Fig 3. Model accuracy from FS-PLS and competing methods on multiclass datasets where each class is assessed separately.** Predictions from test folds within each method are concatenated to provide an overall measure of prediction

accuracy for each class. For both RAPIDS (**A**) and Álvez (**B**) datasets, model accuracy from both unweighted (left) and weighted models (right) are shown, and performance is presented in AUC, Sensitivity, and Specificity per class. For Álvez data, unweighted FS-PLS models were trained through to a maximum number of 10 features regardless of training performance because heavy class imbalance led to deterioration of training loss with the inclusion of any more than one variable. Overall, prediction accuracy from FS-PLS is better in unweighted than weighted models, and vice versa for the competing methods. Hence, our continued discussion of FS-PLS's multiclass performance refers to unweighted FS-PLS models in comparison with the weighted versions derived from LASSO, Elastic-Net and mRMR. **B.** Cumulative probability curves for sample prediction of each class in the two multi-class datasets from FS-PLS's unweighted and mRMR's weighted model. FS-PLS models maintain much greater separation between classes than mRMR as indicated by the area between coloured lines. Lines are coloured by case or control where cases are samples with true class corresponding to the named label of the column and controls are samples from the other classes. Curves are annotated with Wasserstein distance between controls and cases, where higher values indicate more confidence separation of the two by the model.

sample sizes, 20 replicates within each) across the spectrum of sample sizes that were then evaluated on the held-out test partition.

The modelling results (Fig 2) demonstrate that at low sample numbers, models built with ordinary FS perform better than FS-PLS. With increasing sample size, FS-PLS accuracy improves more rapidly than ordinary FS and ultimately outperforms it at mid-high sample sizes. With such limited sample numbers in the Golub dataset, the poor performance of FS-PLS is likely due to the performance differential demonstrated in this sampling experiment. For the Kaforou data, FS-PLS's advantage emerges at the inclusion of few samples (~44). For the Ng data, the shift is more gradual. Performance reaches parity again at few samples (~93), but FS-PLS only begins to outperform FS when the modelling has exhausted all available samples. Similarly, the equal results for FS-PLS and Stagewise FS over the full k-fold train-and-test structure in the Ng data are reflective of the samples available for the experiment, and, given the trend displayed here, FS-PLS may continue to improve on the results of FS with more samples included

To further explore the differences between FS-PLS and Stagewise FS models, we examined the correlation structure of the features selected by the two methods on the Kaforou and Ng datasets. Features selected by FS-PLS consistently display lower pairwise correlation than those chosen by Stagewise FS (Fig 2). The overall divergence in features selected also demonstrates how the two methods forge different solution paths despite their conceptual similarities (see Methods).

## FS-PLS outperforms forward selection on multinomial datasets

For multinomial datasets, FS-PLS achieves higher AUC than stepAIC in all classes for both RAPIDS and Álvez data (Fig 2 and S8 and S9 Data). The improved performance of FS-PLS over stepAIC in multinomial datasets make it a promising method for this use as there are few methods available for sparse multinomial feature selection. Furthermore, it is simpler to construct than StepAIC, as the latter is burdened by the need to construct a full linear model using all features before proceeding through the feature selection steps.

## FS-PLS selects features for normalisation that retain accuracy without requiring a true normalisation factor

Next we addressed the problem of library size normalisation. The development of sparse signatures for diagnosis from omics data is generally hampered by the requirement to measure all the molecules in the assay for normalisation of the data (e.g., library size). We demonstrate a novel use of FS-PLS to learn normalising features in the RNA-Seq datasets which scale reliably and strictly with the dataset's library size. By using these features for normalisation,

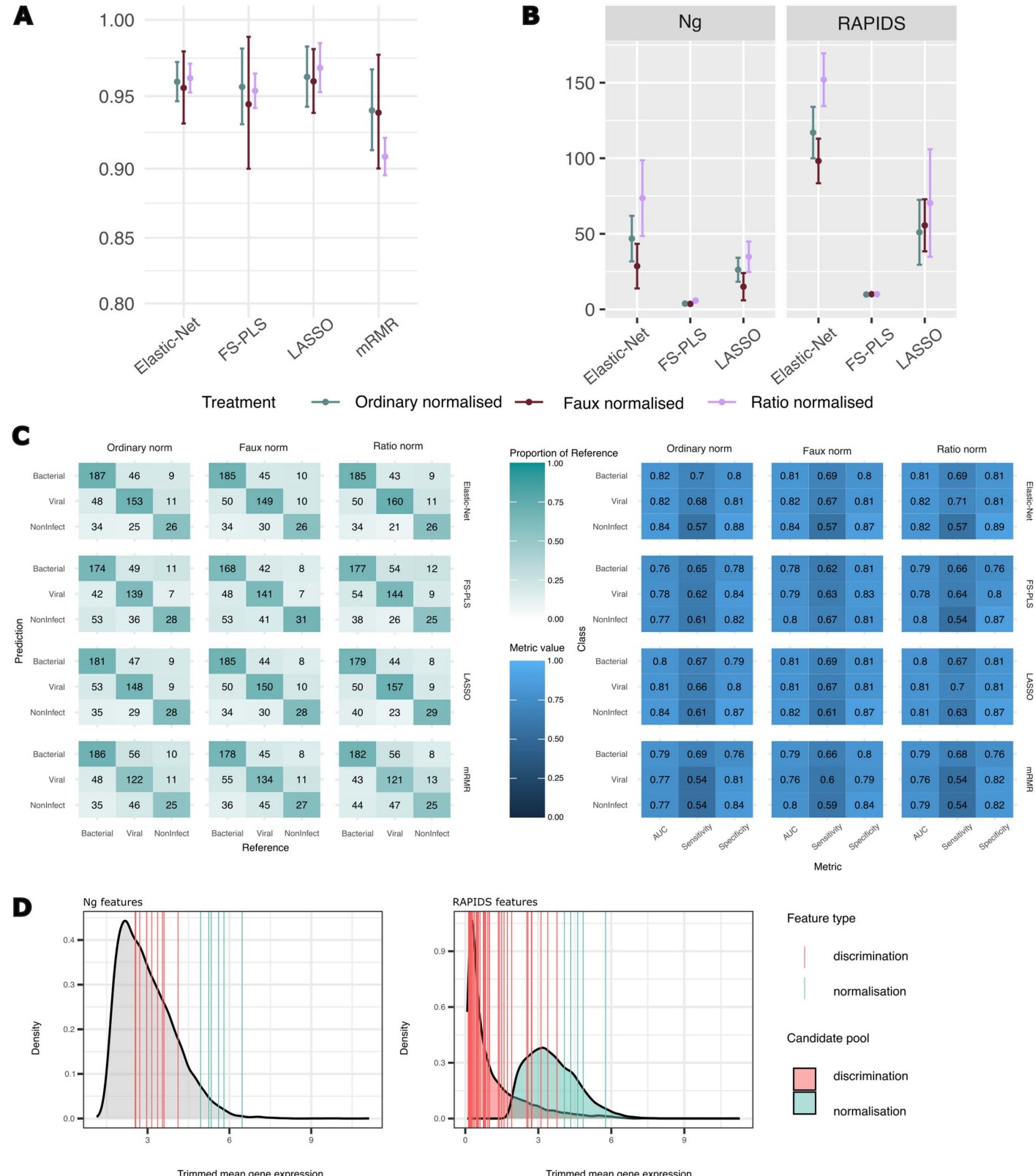

**Fig 4. Results of sparse normalisation approaches on RNA-Seq datasets. A**. Mean and 95%CI AUC results for each fold of Ng data when models constructed using ordinary or alternative normalisation strategies. Normalisation features were selected with FS-PLS before being used to build class discrimination models with

LASSO, Elastic Net, FS-PLS, and mRMR. Broadly, FS-PLS's alternative normalisations produce comparable accuracy results to the full 'ordinary' normalisation. **B.** Numbers of features selected for the models displayed in **A** and **C**. Count excludes the features used in normalisation (n=2 for Faux and Ratio normalisation). As in 2A, number of mRMR features is excluded as it is explicitly instructed to match FS-PLS. What is exchanged for the accuracy differences in **A** and **C** is a complete model with few features needed for normalisation and discrimination of samples. **C.** Concatenated results for class predictions from all 5 folds of RAPIDS data when models constructed on varying normalisation strategies. Left tile plot shows confusion matrix of predictions per method and per normalisation approach with predicted class on the vertical axis. Right plot shows AUC, sensitivity and specificity per class. As above, the compressed 'faux'- and 'ratio'-normalised modelling approaches display only slim accuracy differences while discarding the need for full library size calculation. **D.** Discrimination and normalisation features from FS-PLS Faux normalisation models (5 folds per dataset) displayed according to their trimmed mean expression in the original datasets and superimposed over density plots of the trimmed mean for all features considered. Selected normalisation features tend to be expressed at much higher levels than discrimination features. In the RAPIDS dataset, normalisation and discrimination features were selected from different subsets of the features (10000 highest trimmed mean expressed, and 10000 most variably expressed genes, respectively), so density plots are split according to the different pools. The two feature types were chosen from the same feature subset in the Ng data (10000 highest trimmed mean expressed).

we bypass the need to survey all molecules in the assay and raise the possibility of generating diagnosis signatures that are constituted entirely of a set of discriminating features plus a small selection of features for normalisation of the data.

Using the Ng and RAPIDS datasets, we program FS-PLS to progressively choose features and fit coefficients to reproduce the log library size of each sample until the log-likelihood model of the model is no longer significant against the null model (see *Methods).* A 5-fold structure was used again so that five models were produced for each dataset and performance could be evaluated on the respective hold-out group.

For most datasets and folds, the algorithm halted after two features had been selected and so for simplicity we limited the number of normalising features to two globally. The adjusted $R^2$ value for the correlation between true and predicted log-library sizes in the held-out folds was 0.92 for Ng and 0.88 for RAPIDS (S2 Fig), indicating a high predictive accuracy. We used the selected normalising features in two ways: firstly, to normalise feature counts in each sample by the predicted log-library size, which we call 'Faux' normalisation; and secondly, to generate a new feature matrix constituted of each feature vector divided by the feature vector of each normalisation feature, which we call 'Ratio' normalisation. With 2 normalisation features in a fold, the Ratio normalisation approach leads to a new data matrix of dimension *n* x *2p*. Our implementation of a ratio normalisation was inspired by the work of Wang *et al.* [36] in their Cross-Platform Omics Prediction (CPOP) procedure.

We employed both the alternative normalisation approaches in a new discrimination feature selection round and compared the classification performance of the resulting models against models computed on the ordinarily normalised data. Selection of discriminating features was re-performed with all methods and the same train-test partitions were used for normalisation feature selection to ensure that samples in the held-out folds had never before been encountered by either the normalisation or discrimination model generation.

Fig 4 shows the results of the experiment with alternative normalisation strategies. The Ng data once again delivers similar performance from LASSO, Elastic-Net, FS-PLS and mRMR across the various normalisation strategies. The similar performance is a promising finding, given that the ratio and faux normalisation solutions from FS-PLS now only require an average of 3.6 discrimination features for faux normalisation and 5.8 discrimination features for ratio normalisation, plus two additional normalisation features, to classify a new sample. The result, if replicated in wider cohorts, could present an opportunity for conversion of the omics signature into a point-of-care diagnostics test.

In the RAPIDS dataset, Elastic-Net and LASSO generally outperform the other methods in each normalisation approach, as might be expected given the results that have already been revealed in this study. mRMR and FS-PLS show little difference in accuracy. Once again, however, the great reduction in number of features needed for the model makes the use of FS-PLS

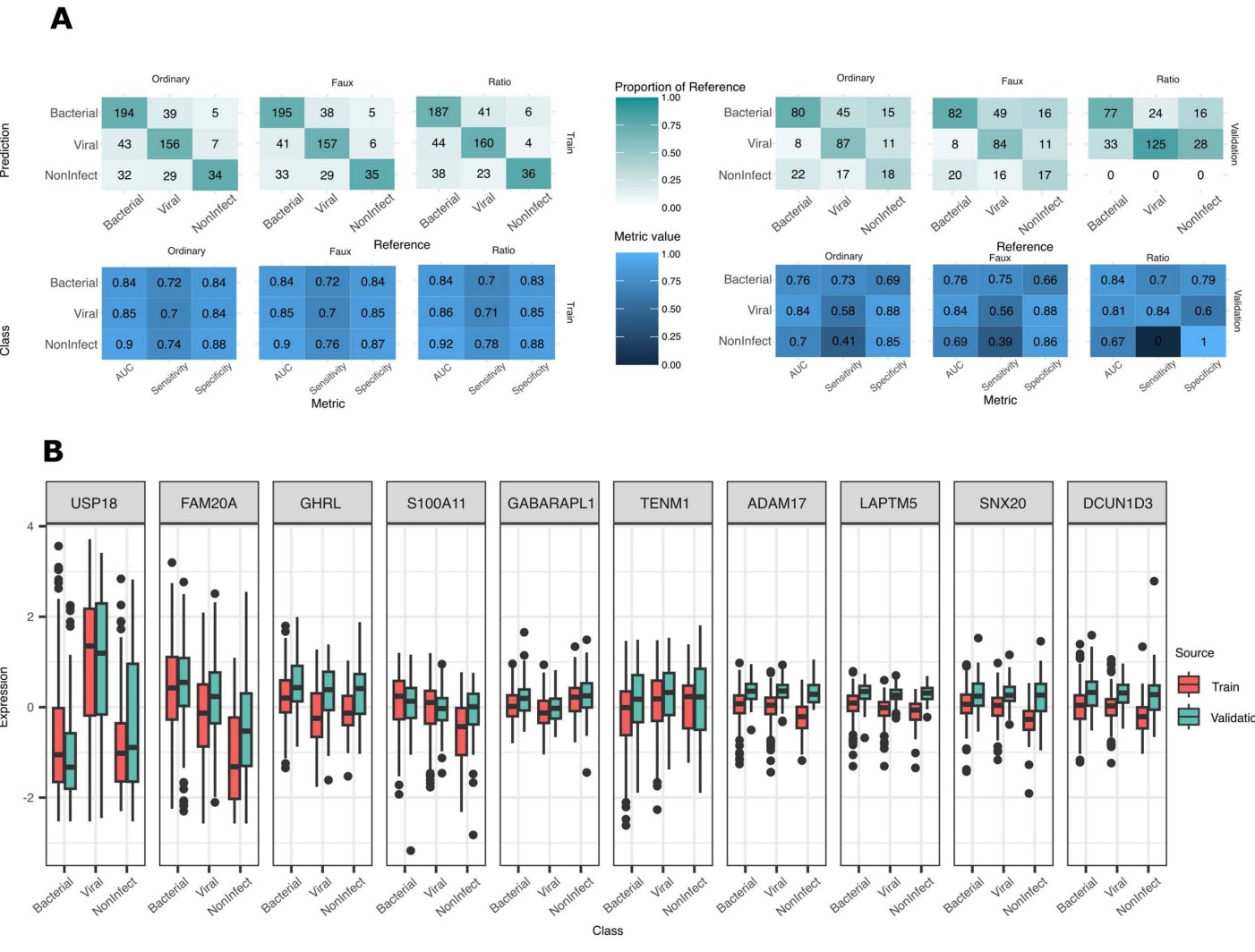

**Fig 5. Performance assessment of FS-PLS on RAPIDS validation data. A.** Performance of FS-PLS models on RAPIDS train and validation data. The left side displays training error, right side displays validation error. The top row of plots are confusion matrices, with each square coloured by the proportion of reference class captured by the prediction. The bottom row of plots displays AUC, Sensitivity, and Specificity metrics for each class. Results are additionally separated according to the normalisation approach used. Model accuracy suffers a sizeable drop in accuracy when applied to validation data and is consistent across normalisation types. **B.** Expression profiles of selected features across training and validation dataset, separated into true class of each sample. Data is from 'ordinary' normalisation approach and shows unmatched distributions across the datasets for many of the chosen features. The drop in validation performance shown in **A** may be related to the shifted distribution of normalised expression values since validation data is centred according to mean feature values learned in the training data.

an attractive prospect for development of a diagnostic test. The total signature size from FS-PLS is 10 discrimination features plus 2 normalisation features on each fold, compared with an average of 50 (plus 2 normalisation features) for LASSO. The 5-fold reduction in feature numbers comes at a small AUC cost of 0.02 in both alternative normalisation models (0.81 multiclass AUC LASSO, 0.79 multiclass AUC FS-PLS).

## Performance of alternatively-normalised models on validation data mirrors performance of ordinary-normalised dataset

To further evaluate the use of FS-PLS for coupled normalisation and classification, we trained FS-PLS on the entire RAPIDS dataset and tested its performance on an external validation set

(Fig 5). As with the cross-validation setup described above, we used FS-PLS to select features predictive of sample log-library size, then proceeded with training FS-PLS to predict diagnosis class from ordinary, faux, or ratio normalised RNA-seq data. All models selected a set of 12 features: 10 for discrimination and 2 for normalisation.

Prediction results on the validation data suffered a severe drop in fidelity (Fig 5), primarily due to a reduced performance on new Non-Infectious samples (0.9 train AUC vs 0.7 validation AUC; Ordinary normalisation). Recovery of the Non-Infectious class has been a difficult task in earlier studies on sepsis biomarkers [37] and, indeed, the ratio-normalised model mis-classified every Non-Infectious sample as Bacterial or Viral. Yet the decline in performance from Training to Validation data is consistent across the Ordinary and Faux normalised models, suggesting that the elimination of the full library normalisation does not inject any further error into the model. Furthermore, models trained with LASSO and Elastic-Net on the three alternatively-normalised datasets also show comparable validation performance with FS-PLS (Elastic-Net Ordinary-normalisation validation AUCs: 0.82 (Bacterial), 0.83 (Viral), 0.69 (Non-Infectious), S3 Fig and S10 Data), and the loss of validation accuracy is more likely attributable to the broader issue of generalisability across experimental batches (Fig 5, see *Discussion*). Since the expanded feature set used in these models might be expected to deliver close to maximum accuracy, the close performance from FS-PLS is a positive result that demonstrates the feasibility of the truly sparse FS-PLS model.

## Discussion

In this work we have benchmarked and formally described an effective method for biomarker discovery, FS-PLS, which derives small predictive signatures of disease. We have demonstrated the flexibility and applicability of the method using five publicly available datasets across three 'omics technologies. In four out of five datasets, FS-PLS produces signatures with a small number of features and high predictive performance in comparison with widely used alternatives.

FS-PLS shares some conceptual similarities with forward stagewise selection. However, we have shown that the variables selected are different, and that forward stagewise selection tends to select variables with much higher pairwise correlation. One important difference between the methods is that forward stagewise selection refits the coefficients of variables which have already been selected at each step, whereas FS-PLS only fits the orthogonal component of the new variables. Another difference is that because FS-PLS does not adjust the outcome variable y, it can continue to use the same link function for each iteration (such as a logistic link function for binary outcome). Conversely, stagewise-FS must use a continuous link function as the residual of a binary outcome variable will no longer be binary.

In cases where FS-PLS models fail to match the performance of other methods, our investigations suggest that small sample numbers are responsible for the disparity. Increasing the number of samples available for training elicits best performance from FS-PLS and enables it to provide accurate predictions from few features. In any case, efforts to develop robust biomarker signatures for diagnosis ought to make use of large training cohorts regardless of the method [38].

FS-PLS will be useful for researchers wishing to derive predictive signatures of disease that require as few features as possible. We anticipate the accelerated development of point-of-care lab-on-a-chip (LOC) diagnostic devices in the coming decade, for which minimal signatures of disease are required. In such a scenario, FS-PLS is well placed to deliver features for conversion into an LOC. We have demonstrated that signatures generated with FS-PLS underperform those from LASSO and Elastic-Net in cross-fold validation. However, the trade-off of size to performance is one that will prompt researchers to use FS-PLS over alternatives in applications that require a small signature.

A major benefit of small diagnostic signatures is that they retain clinical interpretability. We include a summary of the ten features selected by FS-PLS in the RAPIDS ordinary-normalisation signature, including their biological roles, as an illustration of the signature's interpretability (S1 Table). It must be emphasised, however, that the features selected by FS-PLS are not suggested as causative in nature. Rather, they are the features most stably correlated with the outcome and whether causation is involved is a task for another investigation.

A second benefit of small signatures is that they enable conversion of the test into a cheaper assay. qPCR is a cost-effective and widely used diagnostic technology which at present has the most readily converted technology for adoption of small signatures. We report mean FS-PLS signature sizes of 2-10 features from the datasets studied here. Modern qPCR instruments can detect the lower end of the range (generally up to 6 analytes) with multiplexing [39]. Signature sizes at the higher end are still out of reach of multiplexed qPCR and will require continued innovation in the detection technology [40,41]. However, the sizes of signatures output from LASSO are an order of magnitude beyond the capacity of the most advanced qPCR technologies, even though they offer improved diagnostic accuracy. The trade-off between accuracy and signature size is thus not a steady one, but one that massively rewards smaller signature sizes once the signature size overlaps with the number of detectable analytes.

In this study we also demonstrate the usefulness of FS-PLS for selecting features which act to normalise the measurement of the discrimination features. It remains unknown whether normalisation features detected from a high-throughput assay will also prove effective in a condensed LOC test. It may be that such features reflect not a biological control for variation but rather a technical artefact of the assay, in which case their suitability in an alternative technology will be muted. Alternatively, Oxford Nanopore Technologies' targeted sequencing shows potential for miniaturisation and diagnostic specialisation [42,43]. With a technology such as this that has consistent mode of operation in both full and reduced form, determining a normalisation feature using a full assay may only require re-evaluation of the normalisation weights for it to be applied in the specialised application. However, existing demonstrations of rapid targeted sequencing tests have searched for sequence variations (SNPs, structural variants) or binary presence/absence of a genetic feature rather than exploring quantitative gene signatures [44,45].

There are several possible ways the work presented here may be improved. Firstly, the generalisability of FS-PLS models to new data needs to be demonstrated further. There is difficulty in obtaining separate discovery and validation datasets that follow identical sample and library processing steps. When validating our RAPIDS dataset, we aimed to test the model on an earlier dataset [10], but found that variations in the protocol led to poor agreement of the feature expression in the two sets. Likewise, an attempted test of our normalisation strategy using a library model built on an external dataset [46] failed to reproduce an accurate library size in the Ng data because of a ribosomal RNA depletion step in the library preparation. A more robust demonstration of FS-PLS's end-to-end signature generation should use a larger number of datasets with more aligned discovery and validation protocols.

Aside from protocol differences, it is likely that ordinary batch effects are also contributing to the validation performance drop off seen in the RAPIDS data. Our approach intends to create fully online-capable signatures – processing samples individually without any information gleaned from a wider sample set. As such, each feature is centralised by the mean of the feature learned from the training data, which for many features fails to truly centralise the validation data (Fig 5C). Training FS-PLS on a larger set of data from more sources will help to minimise the influence of dataset-specific patterns and produce more generalisable signatures. However, the question of how to reliably scale individual samples is an open one without a clear solution.

We have identified several computational improvements that could be made to the algorithm. Firstly, the program would benefit from a more efficient method of selecting the next most important feature. Least Angle Regression (LARS) [32] is one such method, and in development tests was effective at speeding up analysis of binary datasets. However, there is no trivial way to extend LARS to multinomial data, so in this setting we are at present limited to the relatively slow procedure of fitting a multinomial linear model for every variable at every selection step. Earlier research has demonstrated some potential for the implementation of LARS in a multinomial scenario by using a non-canonical link function [47]. Future research on FS-PLS should investigate ways by which LARS might be extended or approximated for multinomial outcomes.

Modellers of biological data often seek to predict samples on an ordinal scale like disease severity (e.g., mild, moderate, severe) or progression (e.g., early, middle, late). For these outcomes it is clear there is an order to the categories. To make FS-PLS useful for these types of data, existing procedures for feature selection in ordinal data should be investigated for adaptation into our method [48,49]. On this, the advantage in the design of FS-PLS when incorporating new methods is that no adjustments are made to the $y$ vector at each feature inclusion. So long as there are means by which to select individually relevant features, the process can be repeated to include more features after each successive deflation of the $X$ matrix. Indeed, the deflation of $X$ gives FS-PLS a significant advantage over ordinary FS methods for multinomial problems as it avoids the requirement to subtract the estimate $\check{y}$ at each iteration which cannot be performed for categorical data. For this reason, feature selection on multinomial data is achieved with alternative methodologies like *stepAIC* which requires modelling of the full feature space and, as we show above, is outperformed by FS-PLS.

Finally, FS-PLS may benefit from alternative options for stopping criteria. At present, the algorithm stops when the log likelihood of additional features ceases to achieve significant difference from the null model. The criteria permit a new feature to enter the model that is statistically significant even when the additional accuracy it delivers is low. Alternatively, instituting a cross-fold error stopping criteria would allow the algorithm to halt its search when the added feature no longer improves performance in held out samples, thus leading to signatures that increase in size only if they benefit accuracy.

## Supporting information

**S1 Fig. Cumulative probability curves for each disease class from weighted and unweighted models on Álvez and RAPIDS datasets** . See Fig 3B caption for description of plots.
(TIFF)

**S2 Fig. True vs fitted log-library sizes of Ng and RAPIDS datasets where fitted values are given by FS-PLS normalisation models.** Regression line is displayed in red.
(TIFF)

**S3 Fig. Cumulative probability curves for all classes in Discovery and Validation cohorts of RAPIDS data, where probabilities are given by FS-PLS models trained on discovery cohort** . See Fig 3B caption for description of plots.
(TIFF)

**S4 Fig. Correlations of features chosen by Stagewise Forward Selection and FS-PLS on all folds of Kaforou dataset.**
(TIFF)

**S5 Fig.** As with S4 Fig but for Ng dataset.

**S1 Table.** Summary of the ten features selected by FS-PLS in the RAPIDS ordinary-normalisation signature, including their biological roles.
(XLSX)

**S1 Data.** Per-fold AUC accuracy of all models on binary datasets.
(XLSX)

**S2 Data.** Per-fold number of features selected by all models on binary datasets.
(XLSX)

**S3 Data.** Per-fold number of features on RAPIDS dataset, weighted and unweighted models.
(XLSX)

**S4 Data.** Sensitivity, Specificity, and AUC per class from all methods on RAPIDS dataset.
(XLSX)

**S5 Data.** Per-fold number of features on Álvez dataset, weighted and unweighted models.
(XLSX)

**S6 Data.** Sensitivity, Specificity, and AUC per class from all methods on Álvez dataset.
(XLSX)

**S7 Data.** Per-fold AUC accuracy of FS-PLS vs Stagewise-FS on binary datasets.
(XLSX)

**S8 Data.** Sensitivity, Specificity, and AUC accuracy per class of FS-PLS vs StepAIC on RAPIDS dataset.
(XLSX)

**S9 Data.** Sensitivity, Specificity, and AUC accuracy per class of FS-PLS vs StepAIC on Álvez dataset.
(XLSX)

**S10 Data.** Train and validation AUC accuracy per class of all methods and alternative normalisations on RAPIDS dataset, where models were fully trained on all data in the train partition.
(XLSX)

**S1 Info.** Pseudocode of FS-PLS algorithm.
(PDF)

## Acknowledgments

This research was supported by The University of Melbourne's Research Computing Services and the Petascale Campus Initiative. D.R. acknowledges support from the Australian Government Research Training Programme (RTP) scholarship. We acknowledge the assistance of Dr Harrieta Eleftherohorinou in evaluating FS-PLS. We acknowledge all investigators of the Rapid Paediatric Infection Diagnosis in Sepsis (RAPIDS) Study Group.

Rapid Paediatric Infection Diagnosis in Sepsis (RAPIDS) study group:

Luregn J. Schlapbach[7,9,#], Sainath Raman[7,10], Natalie Sharp[7], Natalie Phillips[7,11], Adam Irwin[12,13], Ross Balch[13], Amanda Harley[13], Kerry Johnson[13], Zoe Server[13], Shane George[7,14,15], Keith Grimwood[15,16], Peter J. Snelling[14,15], Arjun Chavan[17], Eleanor Kitkatt[17], Luke Lawton[17], Allison Hempenstall[18], Pelista Pilot[18], Kristen S. Gibbons[7], Renate Le Marsney[7], Antje

Blumenthal[8], Carolyn Pardo[7], Jessica Kling[7], Stephen McPherson[7], Anna D. McDonald[7], Seweryn Bialasiewicz[7], Trang Pham[7], Devika Ganesamoorthy[7], Lachlan Coin[1,6]

[7] Children's Intensive Care Research Program, Child Health Research Centre, The University of Queensland, Brisbane, Australia

[8] Frazer Institute, The University of Queensland, Brisbane, Australia

[9] Department of Intensive Care and Neonatology, and Children's Research Center, University Children`s Hospital Zurich, University of Zurich, Zurich, Switzerland

[10] Paediatric Intensive Care Unit, Queensland Children's Hospital, Brisbane, Australia

[11] Emergency Department, Queensland Children's Hospital, Children's Health Queensland, Brisbane, Australia

[12] Faculty of Medicine, UQ Centre for Clinical Research, University of Queensland, Brisbane, Australia

[13] Infection Management and Prevention Services, Queensland Children's Hospital, Children's Health Queensland, Brisbane, Australia

[14] Department of Emergency Medicine, Gold Coast University Hospital, Southport, Australia

[15] School of Medicine and Dentistry, Griffith University, Southport, Australia

[16] Department of Infectious Disease and Paediatrics, Gold Coast Health, Southport, Australia

[17] Paediatric Intensive Care Unit, Townsville University Hospital, Townsville, Australia

[18] Thursday Island Base Hospital, Thursday Island, Australia

# Representative: luregn.schlapbach@kispi.uzh.ch

## Author contributions

**Conceptualization:** Myrsini Kaforou, Kim-Anh Lê Cao, Lachlan Coin.

**Data curation:** Daniel Rawlinson.

**Formal analysis:** Daniel Rawlinson, Chenxi Zhou.

**Funding acquisition:** Lachlan Coin, RAPIDS Study Group.

**Investigation:** Daniel Rawlinson, Chenxi Zhou, Lachlan Coin.

**Methodology:** Daniel Rawlinson, Chenxi Zhou, Lachlan Coin.

**Project administration:** Lachlan Coin.

**Software:** Daniel Rawlinson, Chenxi Zhou, Lachlan Coin.

**Supervision:** Kim-Anh Lê Cao, Lachlan Coin.

**Validation:** Daniel Rawlinson, Myrsini Kaforou.

**Writing – original draft:** Daniel Rawlinson, Kim-Anh Lê Cao, Lachlan Coin.

**Writing – review & editing:** Daniel Rawlinson, Myrsini Kaforou, Kim-Anh Lê Cao, Lachlan Coin.

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
