## [Decision Letter · Decision Letter 0]

13 Sep 2024

PDIG-D-24-00285

A flexible framework for minimal biomarker signature discovery from clinical omics studies without library size normalisation

PLOS Digital Health

Dear Dr. Coin,

Thank you for submitting your manuscript to PLOS Digital Health. After careful consideration, we feel that it has merit but does not fully meet PLOS Digital Health's publication criteria as it currently stands. Therefore, we invite you to submit a revised version of the manuscript that addresses the points raised during the review process.

Please submit your revised manuscript within 60 days Nov 12 2024 11:59PM. If you will need more time than this to complete your revisions, please reply to this message or contact the journal office at digitalhealth@plos.org. Please include the following items when submitting your revised manuscript:

We look forward to receiving your revised manuscript.

Kind regards,

Hisham Hasan

Guest Editor

PLOS Digital Health

Journal Requirements:

1. We ask that a manuscript source file is provided at Revision. Please upload your manuscript file as a .doc, .docx, .rtf or .tex.

2. We have amended your Competing Interest statement to comply with journal style. We kindly ask that you double check the statement and let us know if anything is incorrect. 

Additional Editor Comments (if provided):

Please follow the author guideline and the journal requirements: http://journals.plos.org/digitalhealth/s/submission-guidelines

Provide more discussion on the biological interpretability of the selected features.

Please address the generalisability concerns by including additional validation datasets and aligning protocols.

Consider exploring more efficient algorithms or techniques to enhance computational efficiency, particularly for multinomial data.

Adapt FS-PLS for ordinal data and explore stopping criteria improvements to ensure that the model selection process is optimized for performance.

Reviewers' comments:

Reviewer's Responses to Questions

**Comments to the Author**

1. Does this manuscript meet PLOS Digital Health’s publication criteria ? Is the manuscript technically sound, and do the data support the conclusions? The manuscript must describe methodologically and ethically rigorous research with conclusions that are appropriately drawn based on the data presented.

Reviewer #1: Yes

Reviewer #2: Yes

2. Has the statistical analysis been performed appropriately and rigorously?

Reviewer #1: N/A

Reviewer #2: Yes

3. Have the authors made all data underlying the findings in their manuscript fully available (please refer to the Data Availability Statement at the start of the manuscript PDF file)?

Reviewer #1: Yes

Reviewer #2: Yes

4. Is the manuscript presented in an intelligible fashion and written in standard English?

PLOS Digital Health does not copyedit accepted manuscripts, so the language in submitted articles must be clear, correct, and unambiguous. Any typographical or grammatical errors should be corrected at revision, so please note any specific errors here.

Reviewer #1: Yes

Reviewer #2: Yes

5. Review Comments to the Author

Please use the space provided to explain your answers to the questions above. You may also include additional comments for the author, including concerns about dual publication, research ethics, or publication ethics. (Please upload your review as an attachment if it exceeds 20,000 characters)

Reviewer #1: Dear Authors,

Thank you for your submission to PLOS Digital Health. I have reviewed your manuscript titled "A flexible framework for minimal biomarker signature discovery from clinical omics studies without library size normalisation." Below are my detailed comments:

Areas for Improvement:

1- The generalizability of the method to be applied: Due to its declining performance in the Golub dataset, concerns have been raised about the effectiveness of FSPLS with respect to smaller sample sizes. In these cases, I propose that further validation be carried out using larger datasets or that a discussion be held on how to adapt or improve FSPLS.

2- Trade vs Signature Size and Performance: While FSPLS is successfully reducing the number of features, it is at a slight cost to predict performance compared to LASSOPLNet and ElasticNet. Consider more detailed discussion of the clinical implications of this transaction, especially in environments where there may be a small reduction in accuracy.

3- Clarity: The manuscript is technically detailed, which may be difficult for a nonexpert to follow. In order to enable the work to be made available to a wider audience, it is appropriate to simplify some of the explanations and provide more detail where necessary.

Reviewer #2: In this work, the authors proposed the forward selection-partial least squares algorithm (FSP-LS) to select a small number of features to classify phenotypes from the high-dimensional omics data. The algorithm iteratively (1) selects one feature that best regresses the phenotype against the residual matrix of the data, (2) adds the newly selected feature to the existing selected feature set, and projects the data onto the subspace spanned by the updated feature set, (3) subtracts the projection from the data and generates a residual matrix. They compared the performance of FS-PLS with five other methods on five transcriptomic datasets. Overall, FS-PLS selects much fewer biomarkers than other methods but still retains similar prediction accuracy. It also selects features for normalization without requiring a normalization factor.

The ideas of the proposed method are well presented, and the results look convincing. Yet I have concerns on several major issues.

First, as the authors point out, FS-PLS and Stagewise-FS yield similar performance, and both methods also return a small number of biomarkers in the empirical data analysis. Intuitively, at each iteration Stagewise-FS should also select a new feature which is independent of existing features and best fits the residual of the phenotype. A candidate feature which is a linear combinations of existing features will not fit the residual well since its explanatory power is absorbed by the existing features. The authors should provide the reasons or intuitive explanations why FS-PLS is superior to Stagewise-FS.

Second, a major benefit of FS-PLS claimed by the authors is that it returns a small number of features which yield similar prediction accuracy to other methods. This is important but not sufficient to prove the utility of selected biomarkers. Biomedical experts often demand the biological meanings of the selected markers, especially when FS-PLS returns a very small number of markers. In the analysis of the experimental datasets, what are the selected marker genes? Do they have any empirical/biological support for phenotype prediction? How can one address the biological meaning of the selected markers?

Third, the way the authors handle projections and residuals is not very obvious to me. The projection P^{k} has the same dimension as the original data matrix X. This seems to contradict with my intuition that a projection maps the original data into a lower dimensional space. To me an obvious procedure with the spirit of the proposed algorithm is as follows. At each iteration keep the candidate features which are orthogonal to all the selected features. Regress the phenotype against the selected features and get the residuals. Regress the residuals against each candidate feature, and select the candidate feature that best fits the residual. What's the difference between this obvious procedure (I think it's a wrapper method) and FS-PLS (a combination of filter and embedding approaches)? More fundamentally, if the projection is linear (like the current method), then why bother applying the projection rather than sequentially adding features?

Fourth, it is not clear what is the "univariate linear model" at step 1 of the algorithm when the output variable is categorical (e.g., ALL vs AML). Will it be better to leverage the power of nonlinear functions (such as SVM and NN) to the multi-dimensional data of selected features rather than using linear models to single features?

6. PLOS authors have the option to publish the peer review history of their article (what does this mean? ). If published, this will include your full peer review and any attached files.

**Do you want your identity to be public for this peer review?** For information about this choice, including consent withdrawal, please see our Privacy Policy .

Reviewer #1: Yes: Nadeem M. Salman

Reviewer #2: No

---

## [Decision Letter · Decision Letter 1]

10 Feb 2025

A flexible framework for minimal biomarker signature discovery from clinical omics studies without library size normalisation

PDIG-D-24-00285R1

Dear Dr. Coin,

We are pleased to inform you that your manuscript 'A flexible framework for minimal biomarker signature discovery from clinical omics studies without library size normalisation' has been provisionally accepted for publication in PLOS Digital Health.

Best regards,

Hisham E. Hasan

Guest Editor

PLOS Digital Health

**Additional Editor Comments (if provided):**

There are a few minor typographical and grammatical issues in the manuscript (e.g., "regualrisation" should be "regularisation," "rela(ve" should be "relative"). A final proofread for consistency and clarity will improve the readability.

There are a few minor formatting issues with the references (e.g., missing page numbers, inconsistent journal formatting). These should be reviewed for consistency with the journal's guidelines.

Some sections mention Supplementary Data (e.g., Supplementary Data 3-6, Supplementary Figure 1), but these are not provided in the main text or seem to be referenced inconsistently. Make sure these references are consistent.

**Reviewer Comments (if any, and for reference):**

Reviewer's Responses to Questions

**Comments to the Author**

1. If the authors have adequately addressed your comments raised in a previous round of review and you feel that this manuscript is now acceptable for publication, you may indicate that here to bypass the “Comments to the Author” section, enter your conflict of interest statement in the “Confidential to Editor” section, and submit your "Accept" recommendation.

Reviewer #2: All comments have been addressed

Reviewer #3: (No Response)

2. Does this manuscript meet PLOS Digital Health’s publication criteria ? Is the manuscript technically sound, and do the data support the conclusions? The manuscript must describe methodologically and ethically rigorous research with conclusions that are appropriately drawn based on the data presented.

Reviewer #2: Yes

Reviewer #3: (No Response)

3. Has the statistical analysis been performed appropriately and rigorously?

Reviewer #2: Yes

Reviewer #3: (No Response)

4. Have the authors made all data underlying the findings in their manuscript fully available (please refer to the Data Availability Statement at the start of the manuscript PDF file)?

Reviewer #2: Yes

Reviewer #3: (No Response)

5. Is the manuscript presented in an intelligible fashion and written in standard English?

Reviewer #2: Yes

Reviewer #3: (No Response)

6. Review Comments to the Author

Reviewer #2: The authors have addressed all my comments in the revised manuscript.

Reviewer #3: The authors responded positively to concerns.

7. PLOS authors have the option to publish the peer review history of their article (what does this mean? ). If published, this will include your full peer review and any attached files.

**Do you want your identity to be public for this peer review?** For information about this choice, including consent withdrawal, please see our Privacy Policy .

Reviewer #2: No

Reviewer #3: None
